# Identification of spatially variable genes with graph cuts

Ke Zhang[1,4], Wanwan Feng[1,4] & Peng Wang ●[1,2,3] ✉

Single-cell gene expression data with positional information is critical to dissect mechanisms and architectures of multicellular organisms, but the potential is limited by the scalability of current data analysis strategies. Here, we present scGCO, a method based on fast optimization of hidden Markov Random Fields with graph cuts to identify spatially variable genes. Comparing to existing methods, scGCO delivers a superior performance with lower false positive rate and improved specificity, while demonstrates a more robust performance in the presence of noises. Critically, scGCO scales near linearly with inputs and demonstrates orders of magnitude better running time and memory requirement than existing methods, and could represent a valuable solution when spatial transcriptomics data grows into millions of data points and beyond.

Systematic assessment of the spatial context of gene expression is a cornerstone in understanding the mechanistic functionality and molecular organization of tissues and organs[1,2]. Currently, two main classes of experimental approaches have been established. Utilizing probes for individual RNA molecules to directly quantify gene expression in situ, image-based single-cell spatial transcriptomics, such as seqFISH[3] and MERFISH[4], can measure hundreds of genes in an entire tissue section with single-cell resolution. Alternatively, by combining RNA-Seq with positional barcoding, genome-scale spatial transcriptomics can be generated for hundreds of tiny spots each containing multiple cells[5,6].

A basic task of analyzing spatial transcriptomics data is to identify spatially variable (SV) genes: here defined as genes whose expression distributions display significant dependence on their spatial locations. Besides the statistical characteristics, recent transcriptome-wide studies indicated that SV genes could also demonstrate a strong conservation in their spatial patterns, such that many SV genes display similar dependencies on spatial locations, resulting in similar trends in spatial patterns of their expression values[5]. Furthermore, SV genes are often markers or essential regulators for tissue pattern formation and homeostasis, consequently, the expression patterns of SV genes generally align remarkably well with underlying tissue structures. Recently, three prominent methods based on marked point process (trendSceek)[7], Gaussian process (spatialDE)[8], or Generalized linear spatial model (SPARK)[9] were developed to identify SV genes. Although these methods have been shown to identify SV genes successfully, these algorithms have computational efficiency of $O(n^2)$ or $O(n^3)$[7–9], limiting their utilities as spatial transcriptomics data grows into millions of data points and beyond.

Here, we present a scalable algorithm, single-cell graph cuts optimization (scGCO), to identify SV genes. ScGCO utilizes a hidden Markov random field (HMRF), a probabilistic graph model that captures statistical (conditional independence) and spatial properties of modeled variables, to identify SV genes. To scale the existing graph cuts algorithm to genome-wide analyses, we developed a heuristic method to automatically set smooth factor, the hyperparameter in graph cuts, by optimizing the signal-to-noise ratio (SNR) of graph cuts results. Extensive analyses with both simulated data and multiple spatial transcriptomics datasets from a wide variety of biological samples suggested that scGCO identified one of the highest numbers of SV genes while maintained a better robustness and a more favorable false positive rate (FPR) than existing methods. More importantly, scGCO demonstrated improved scalability and could process millions of data points with a desktop computer. With the number of analyzed data points growing beyond millions in a single experiment, scGCO

[1]National Genomics Data Center, CAS Key Laboratory of Computational Biology, Bio-Med Big Data Center, Shanghai Institute of Nutrition and Health, University of Chinese Academy of Sciences, Chinese Academy of Sciences, Shanghai, China. [2]Faculty of Health Science, University of Macau, Macau, China. [3]Ministry of Education Frontiers Science Center for Precision Oncology, University of Macau, Macau, China. [4]These authors contributed equally: Ke Zhang, Wanwan Feng. ✉e-mail: ethanpwang@um.edu.mo

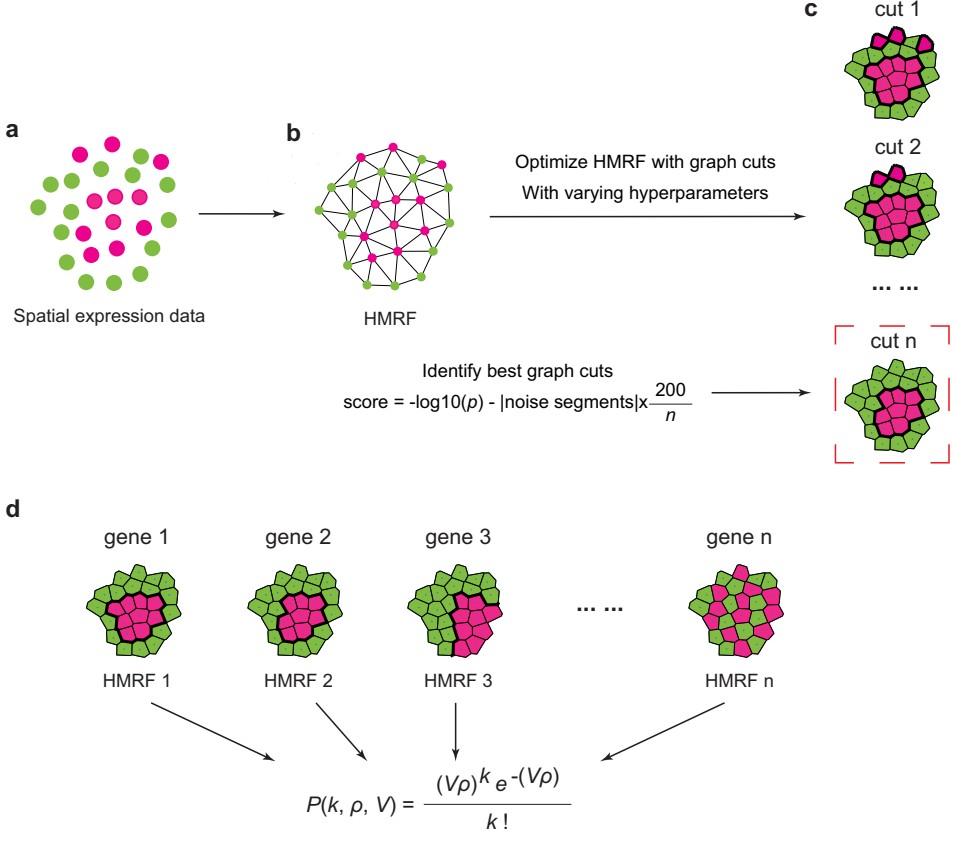

**Fig. 1 | Overview of scGCO for SV gene identification. a** A gene's spatial expression pattern. Each dot represents a cell and is placed according to its spatial coordinate. **b** Representing a gene's spatial expression with hidden Markov random field (HMRF). **c** Optimizing HMRF using graph cuts algorithm with different smooth factors and identifying the best graph cuts result that maximizes a score based on the signal-to-noise ratio. **d** *P*-value for each gene was evaluated using the best segmentation under the complete spatial randomness (CSR) framework. Benjamini–Hochberg (BH) correction was utilized to identify spatially variable (SV) genes at genome-scale. Cells are represented with Voronoi diagrams. Thicker lines highlight the segmentation boundaries identified by graph cuts.

could be a valuable tool to fully realize the potential of single-cell spatial transcriptomics data.

## Results

### Overview of scGCO method

ScGCO models spatial gene expression as a marked point process where points represent the spatial locations of measured cells or spots, and marks are discrete gene expression states (downregulated, upregulated, etc.) associated with points. The dependency of points with a specific mark on spatial locations can then be analyzed under the complete spatial randomness (CSR) framework. The null hypothesis of CSR model assumes that points with a specific mark in a 2D space are distributed in a completely random fashion and can be described by homogeneous spatial Poisson process. Consequently, the probability of observing a certain number of points (cells/spots) with a particular mark (gene expression state) within a specified area can be calculated exactly following homogeneous spatial Poisson process (see methods). For a particular spatial distribution of gene expression, if there are regions whose number of cells/spots of specific marks are associated with statistically significant low probabilities according to the CSR model, we can reject the null hypothesis, and conclude that the gene expression is dependent on spatial locations and designate the gene as spatially variable. Unfortunately, for a spatial distribution of points, the exact locations and shapes of such "spatially dependent" regions are unknown, and previous methods, such as trendSceek[7], have been developed to approximate the CSR model using distributions of pairwise distances between points without explicitly identifying the regions demonstrating spatial dependence. Although these methods demonstrated usefulness in identifying SV genes, they cannot illustrate the exact boundaries of regions demonstrating spatial dependence, which are often of biological interests, and more importantly, these algorithms have computational efficiency of $O(n^2)$ or worse and are unsuitable for analyzing large spatial transcriptomics datasets.

A key advantage of scGCO to overcome these limitations is to utilize HMRF, an effective Bayesian method for object recognition in computer vision, to explicitly identify candidate regions for spatial dependence, which allows scGCO to directly test for spatial dependence under the CSR model. To model the spatial gene expression data with HMRF, scGCO first employs Delaunay triangulation to transform spatial locations of measured cells/spots into an undirected graph, where each node represents a data point (a single cell or a spot measuring multiple cells, depending on the technologies utilized) in the spatial transcriptomics data (Fig. 1). Next, scGCO utilizes Gaussian mixture modeling to separate a gene's expression values into different bins, which represent different gene expression states. The float gene expression value associated with each node in the HMRF was then transformed into corresponding bin number, and the resulting bin number was assigned to the node, creating an initial label assignment for the HMRF. The initialized HMRF was then optimized by the fast graph cuts algorithm of Boykov et al.[10–12] to learn the true hidden labels of the nodes in HMRF, which represent authentic gene expression states. Because the HMRF model penalizes neighboring nodes with different labels, the learned labels optimized by HMRF will naturally form segments containing identical labels. These HMRF-identified

segments presumably represent authentic spatial structures of the analyzed tissues. We then used these segments as the candidate regions to test for spatial dependence of observed gene expression under the CSR model.

A key obstacle in optimizing HMRF is to determine the best value for the hyperparameter: smooth factor in the graph cuts algorithm. To enable efficient optimization of HMRF at genome-scale, we developed a heuristic sequential search procedure to identify the best smooth factor that maximizes the signal-to-noise ratio of graph cuts results, which helps to identify the best segmentation. ScGCO then uses the best segmentations as candidate regions to test whether the observed gene expression states demonstrate significant dependence on spatial locations under the CSR model. Finally, the $p$-values of all genes were adjusted with the Benjamini–Hochberg Procedure to identify statistically significant SV genes at genome-scale.

## Performance evaluation using simulated spatial gene expression data

Because a gold standard SV gene set is not available, we first compared scGCO's performance with existing methods using a simulated dataset consisting of 1000 SV genes and 9000 random genes. Specifically, we simulated ten spatial patterns: 7 artificial patterns representing a wide range of shapes from simple linear strip to complex nonlinear Swiss rolls; and three authentic biological patterns extracted from mouse olfactory bulb tissue structures (Fig. 2a). We simulated 100 samples for each of the ten patterns, creating 1000 SV genes. To generate random samples without spatial dependence, we randomly shuffled the 1000 SV genes by randomly assigning expression values to different nodes while keeping each node's spatial location fixed. This random shuffling process was repeated nine times to create 9000 random samples. Clustering analyses demonstrated that the 1000 SV genes form ten well-separated clusters occupying a broad space when projected into 2D plane via t-distributed stochastic neighbor embedding (t-SNE)[13], suggesting that our simulated patterns demonstrate sufficient complexity and can be used to robustly estimated algorithms' performance (Supplementary Fig. 1a). Expectedly, the 9000 random samples don't form any clusters when projected into 2D plane, confirming that their expression values don't demonstrate spatial dependence, and can be effectively used as negative controls (Supplementary Fig. 1b).

To further evaluate the robustness of tested methods, we added increasing amounts of Gaussian noises into the 10, 000 simulated genes, creating six datasets with different levels of noises. We then ran two published algorithms: SPARK and spatialDE, together with scGCO on these datasets, and evaluated their capacity to identify SV genes (Fig. 2b and Supplementary Fig. 2). At low noise levels, these methods delivered comparable performance. Importantly, scGCO excelled at high noise levels and delivered the best accuracy, sensitivity and F1 score when the Gaussian noise was increased to 0.6. ScGCO also demonstrated a second best false positive rate (FPR), which is well-controlled even at noise level of 0.6, with average FPR of 0.0013 (Fig. 2c).

The excellent performance of tested methods suggested that Gaussian noises are not strong enough to perturb analyzed methods. To further evaluate these methods, we introduced additional perturbations by first randomly selecting some nodes, then randomly exchanging their expression values (Fig. 2d and Supplementary Fig. 3). Under such dramatic perturbations, the performance of all methods deteriorated significantly as the percentage of perturbed nodes increases. Importantly, scGCO delivered the highest robustness to random exchanges and delivered the best performance among all tested methods in terms of accuracy, sensitivity and F1 scores, while maintained a similar FPR (Fig. 2e). Taken together, the comprehensive evaluation using simulated datasets suggested that scGCO is a highly robust method that delivers excellent performance in identifying SV genes.

## ScGCO identifies SV genes from mouse olfactory bulb data

We next applied scGCO to spatial transcriptomics data from mouse olfactory bulb (MOB)[5] measuring gene expression in spots with a diameter of 100 µm, which consists of multiple cells. Because trendSceek only identified < 100 genes in two out of twelve replicates[7], we focused on the comparison with spatialDE and SPARK. We first applied scGCO to replicate 11 of the MOB data, which spatialDE and SPARK analyzed extensively in their study[8,9]. ScGCO identified 796 SV genes (adjusted $p$-value < 0.05), which is about 12-fold more than spatialDE (67 genes) and is comparable to SPARK (772 genes) (Fig. 3a). Extending the analysis to all 12 replicates revealed a similar picture: scGCO (665.00 ± 342.57) on average identified a similar number of SV genes with SPARK (693.50 ± 302.56), which are significantly more than spatialDE (332.83 ± 166.97) (Supplementary Fig. 4, $p$-value < 0.01).

To examine the authenticity of identified SV genes, we next evaluated their biological relevance. Consistent with previous observations that many SV genes share similar spatial patterns[5], SV genes identified by scGCO consistently formed four clusters when projected onto a low-dimensional space via uniform manifold approximation and projection (UMAP)[14] across all 12 replicates (Fig. 3b and Supplementary Fig. 5). Although SV genes identified by spatialDE or SPARK also formed clusters, the number of clusters varied across different replicates, suggesting that they are less robust (Supplementary Figs. 6 and 7). To gain further biological insights into the formed clusters, we examined the spatial expression trends of genes in each cluster and compared them to known MOB tissue structures derived from H&E staining, which consists of five annotated layers (Fig. 3c). Reassuringly, genes identified by scGCO in each cluster demonstrated different trends that aligned with different layers of the known tissue structures (Fig. 3d, e). For example, in replicate 11, genes in cluster 0 are strictly overexpressed in the GCL layer, genes in cluster 1 are overexpressed in ONL layer, genes in cluster 2 are overexpressed in OPL and MCL layers, and genes in cluster 3 are underexpressed in GCL layer (Fig. 3d, e). Expectedly, each cluster of genes can resolve matching tissue structures (Fig. 3d). Similar results were also observed in other replicates, suggesting that they are marker genes for specific spatial domains, and represent authentic SV genes (Supplementary Figs. 8 and 9).

Because a common characteristic of SV genes is the alignment with known tissue structures, we next quantified how well do the identified genes satisfy this criterion. We first examined whether the identified SV genes could resolve all the five known layers of MOB tissue structure with a high accuracy using results from replicate 11 (Fig. 3f, g). To maximize the power to resolve true tissue structures (using more authentic SV genes), while minimizing erroneous structures from false positive SV genes, we used the SV genes commonly identified by the three tested methods as the positive reference. Indeed, all five layers of MOB tissue structures can be resolved using common SV genes (Fig. 3f), with over 90% of the spots in the reconstructed tissue structure can be correctly assigned to the matching layer derived from H&E staining, confirming that the set of common SV genes are valid positive references (Fig. 3f). Importantly, the MOB tissue structure resolved using SV genes uniquely identified by scGCO could also resolve all five layers that matches the known structure well (Fig. 3g). On the contrary, the tissues structures resolved using SPARK-only or spatialDE-only genes are noticeably different from known MOB structures and couldn't resolve all five layers (Fig. 3g).

To examine this issue further, we utilized four metrics, accuracy, precision, sensitivity, and F1 score, to quantify how well could the sets of unique genes resolve MOB tissue structures, using the structure resolved by common genes as the positive reference. SV genes uniquely identified by scGCO delivered the best performance in all four metrics comparing to SV genes uniquely identified by SPARK or spatialDE (Fig. 3h). For example, scGCO-only genes could assign 73.66% of nodes to the correct layers, while SPARK-only and spatialDE-only genes could only assign 52.67% and 16.79% nodes to the correct

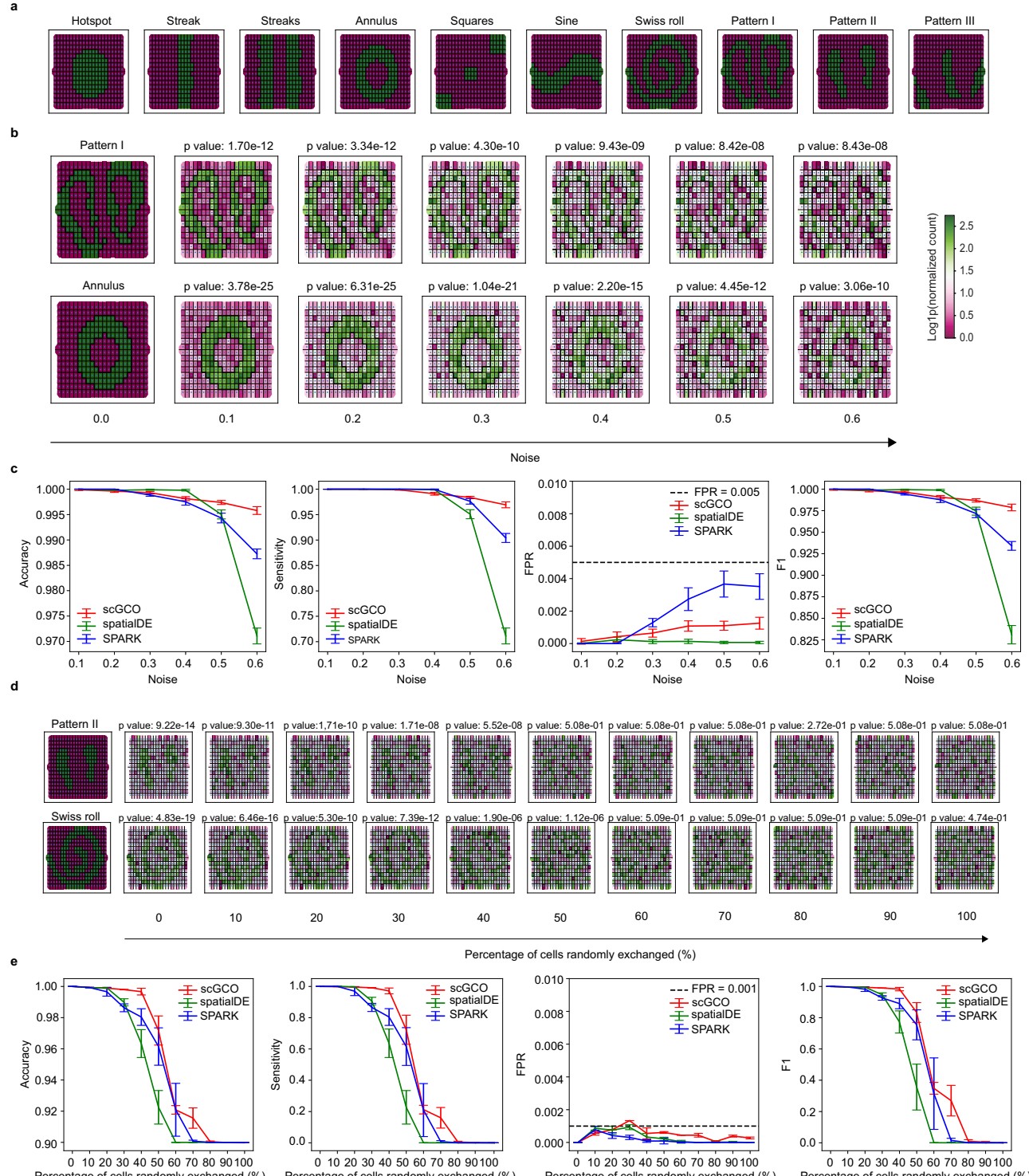

**Fig. 2 | Performance of SV gene identification algorithms using simulated data.** **a** The ten spatial patterns utilized in our simulation. **b** Representative graph cuts results at increasing Gaussian noise levels for scGCO. *P*-values were calculated by scGCO without multiple-testing correction. **c** Line plots showing accuracy, sensitivity, false positive rate (FPR) and F1 score for SPARK, spatialDE, and scGCO with increasing Gaussian noises. Plots were generated from $n = 10$ independent simulations at each noise level. Error bars indicate the means ± SD. **d** Representative graph cuts results at increasing percentage of randomly exchanged cells for scGCO at Gaussian noise level of 0.3. *P*-values were calculated by scGCO without multiple-testing correction. **e** Line plots showing accuracy, sensitivity, FPR and F1 score for SPARK, spatialDE, and scGCO with increasing percentage of randomly exchanged cells. Plots were generated from $n = 10$ independent simulations at each exchange rate. Error bars indicate the means ± SD. Source data are provided as a Source Data file.

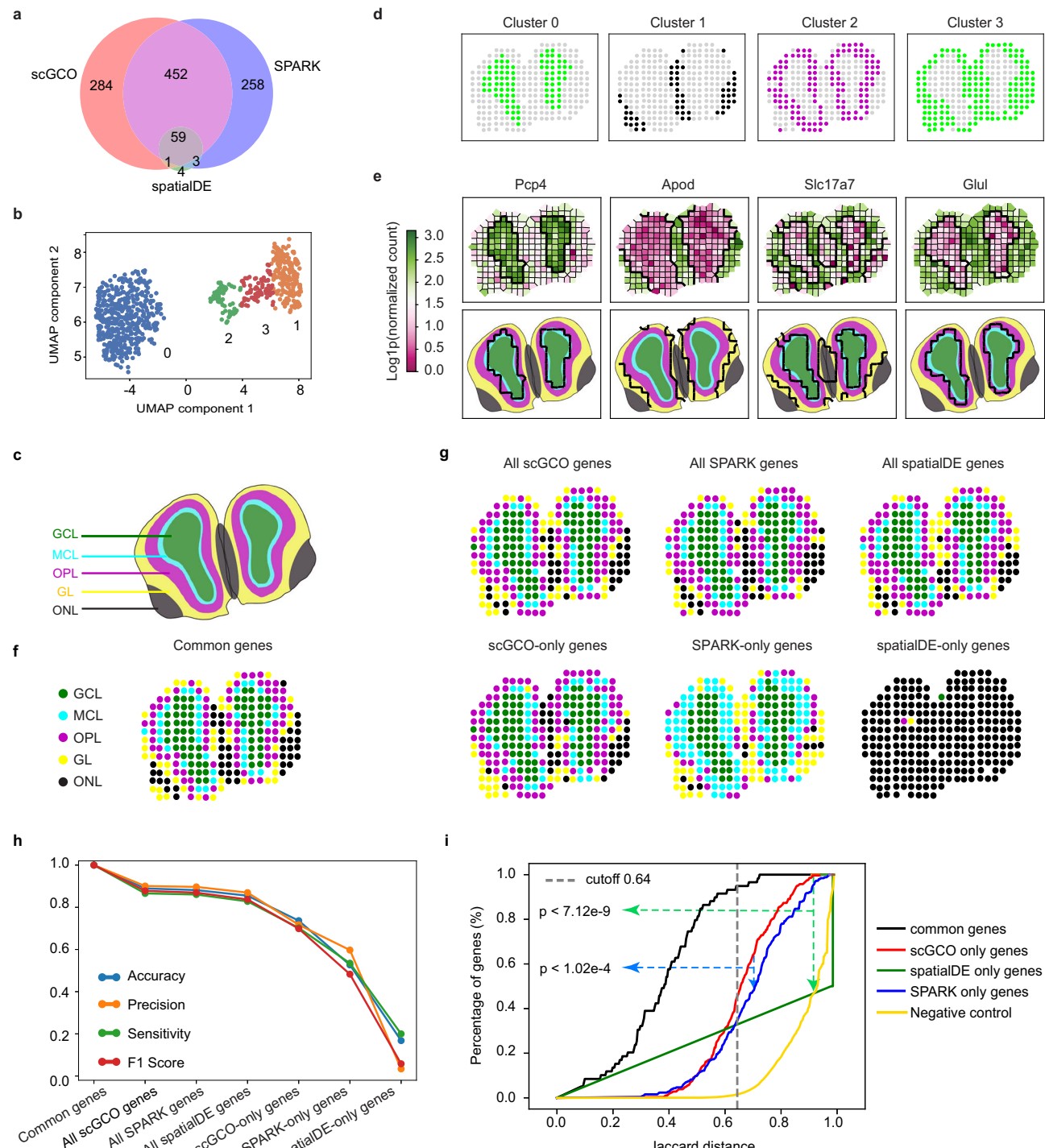

**Fig. 3 | Application of scGCO to spatial gene expression data from mouse olfactory bulb. a** Venn diagram showing the set relationship among SV genes identified by scGCO, spatialDE and SPARK (FDR < 0.05) from replicate 11 of mouse olfactory bulb (MOB) data. **b** UMAP analysis for SV genes identified by scGCO in (**a**). **c** Annotation of the known MOB tissue architecture showing the five layers for replicate 11. GCL: granular cell layer; MCL: mitral cell layer; OPL: outer plexiform layer; GL: glomerular layer; ONL: olfactory nerve layer. **d** Tissue structures resolved with each cluster of SV genes in (**b**). **e** Representative Voronoi diagrams showing graph cuts results (top panels), and tissue structure overlays (lower panels) for genes in (**b**). One example was shown for each cluster. Thicker black lines represent graph cuts boundaries. **f** Reconstructed MOB tissue architecture using SV genes jointly identified by all three methods. **g** The MOB tissue structures resolved using all scGCO genes (upper column 1), all SPARK genes (upper column 2), all spatialDE

genes (upper column 3), scGCO-only genes (lower column 1), SPARK-only genes (lower column 2), and spatialDE-only genes (lower column 3). **h** Line plots showing accuracy, precision, sensitivity and F1 score of the reconstructed tissue structures in (**g**), using the tissue structure reconstructed with common genes as reference. **i** Line plots showing the cumulative distribution of Jaccard distance of identified SV genes to the reference MOB tissue structure reconstructed with common genes for replicate 11. Black line: SV genes commonly identified by the three methods; red line: SV genes uniquely identified by scGCO; green line: SV genes uniquely identified by spatialDE; blue line: SV genes uniquely identified by SPARK; gold line: genes with random expression patterns as negative control. The vertical gray dashed line indicates the estimate distance cutoff corresponding to 95% of common genes. *P*-values were calculated using the two-sided pairwise Kolmogorov–Smirnov (KS) test without multiple-testing correction. Source data are provided as a Source Data file.

layers in replicate 11, respectively. We then extended the analyses to all 12 replicates, and the comprehensive analyses confirmed that SV genes uniquely identified by scGCO on average are best in resolving MOB tissue structures among the tested methods (Supplementary Figs. 10 and 11).

To further quantify the biological relevance of identified SV genes, we next evaluated the similarity of each identified SV gene with the positive reference MOB tissue structures reconstructed with common genes. To this aim, we aligned each gene's spatial expression trend with the reference MOB tissue structures. To achieve a robust evaluation, we used three different metrics: normalized Hamming distance, Jaccard distance, and Hausdorff distance. To further illustrate the significance of observed alignments between putative SV genes and the reference MOB tissue structures, we also calculate these metrics for genes with randomly generate spatial expression patterns, which serve as negative controls. Reassuringly, the commonly identified genes demonstrated the best alignment with the reference tissue structures in all three metrics across the twelve replicates. On the contrary, random genes generally demonstrated the poorest alignments with the reference tissue structures (Fig. 3i and Supplementary Fig. 12). Importantly, SV genes uniquely identified by scGCO consistently demonstrated better alignments with the reference tissue structures than genes uniquely identified by SPARK or spatialDE ($p < 0.0001$ for Hamming distance, Jaccard distance and Hausdorff distance, Kolmogorov–Smirnov test) (Fig. 3i and Supplementary Fig. 12).

To gain a quantitative understanding of the FPR using the cumulative distributions of these metrics, thresholds for false positives were determined for each metric which classified 95% of the commonly identified SV genes as true positives. We then applied the same threshold to genes uniquely identified by each method. For all 12 replicates, SV genes uniquely identified by scGCO consistently demonstrated the lowest FPR in all three metrics using the established thresholds (Supplementary Fig. 12). For example, 45.07% of scGCO-only genes demonstrated a better Jaccard distance than the threshold (0.64) in replicate 11. On the contrary, only 34.90% of SPARK-only genes and 0% of spatialDE-only genes passed the threshold in replicate 11 (Fig. 3i). Furthermore, direct visualization of SV genes uniquely identified by each method confirmed that scGCO-only genes indeed demonstrated more similarities to the reference MOB tissues structures (Supplementary Figs. 13 and 14). Finally, the SV genes identified by scGCO were enriched with neuron-related GO terms and pathways (Supplementary Fig. 15), confirming their biological relevance.

## Methods ignoring spatial context are not effective in identifying SV genes

We next compared SV genes identified by scGCO with genes identified by methods without considering spatial information (Supplementary Fig. 16). Genes identified by scGCO accounted for less than 13.0% (377 of 2894) of highly variable genes (HVGs) identified by Seurat while ignoring spatial context, or less than 26.3 % (473 of 1798) of genes identified by DESeq2 by comparing gene expression in the five layers of MOB tissue against each other in replicate 11. Importantly, most genes uniquely identified by DESeq2 or HVGs demonstrated much worse alignment with the reference MOB tissue structures than SV genes uniquely identified by scGCO (ScGCO vs. DESeq2: $p$-value = 6.39e−48 for Hamming distance, $p$-value = 9.60e−52 for Jaccard distance, and $p$-value = 4.79e−40 for Hausdorff distance. ScGCO vs. HVGs: $p$-value = 1.02e−132 for Hamming distance, $p$-value = 4.06e−137 for Jaccard distance, and $p$-value = 4.52e−85 for Hausdorff distance, determined by Kolmogorov–Smirnov test) (Supplementary Fig. 16b), confirming that HVGs or DESeq2 are not viable methods to identify SV genes. The DESeq2 results revealed that biologically relevant spatial patterns are not required to generate significant differential expressions between different tissue structures (Supplementary Fig. 16c–f),

highlighting the importance to explicitly model spatial variabilities in identifying biologically relevant SV genes.

## ScGCO identifies SV genes from breast cancer data

We next applied scGCO to spatial gene expression data from breast cancer biopsies, which measured four layers of the same sample using identical technologies as the MOB data. The biopsies consist of three types of tissues, invasive ductal cancer (INV), ductal cancer in situ (DC), and normal tissues (NT)[5]. ScGCO identified 309 SV genes in layer 2 (adjusted $p$-value < 0.05), which is about 3-fold more than spatialDE (115 genes) and is comparable with SPARK (290 genes) (Fig. 4a). Extending the analysis to all four layers demonstrated that scGCO identified significantly more SV genes ($237.50 \pm 64.00$) than spatialDE ($100.25 \pm 35.15$), while remained comparable to SPARK ($271.25 \pm 45.42$) across all 4 layers (Supplementary Fig. 17a, b, $p$-value < 0.05). SV genes identified by scGCO and SPARK also consistently formed three clusters across all layers, a phenomenon not observed in random genes or genes identified by spatialDE (Fig. 4b and Supplementary Fig. 17c–f). Expectedly, each cluster of scGCO genes readily recapitulated tissue regions corresponding to INV, DC, and NT tissues, suggesting that scGCO robustly identified marker genes for the three tissue types (Fig. 4c–e and Supplementary Fig. 18a–c).

We next quantified whether SV genes identified by each method could effectively reconstruct the INV, DC and NT regions in underlying cancer tissue, using the structure reconstructed by SV genes commonly identified by the three methods as the positive reference. Expectedly, SV genes uniquely identified by scGCO consistently demonstrated better performance than SV genes uniquely identified by SPARK or spatialDE (Fig. 4f, g, Supplementary Figs. 18 and 19) across all four layers. For example, in layer 2 scGCO-only genes could assign over 65.34% of nodes to the correct regions, while SPARK-only and spatialDE-only genes could only assign 49.80% and 48.20% nodes to the correct regions, respectively. We next evaluated the similarity of each identified SV genes with the reference breast cancer tissue structures reconstructed with commonly identified SV genes. Expectedly, scGCO-only genes consistently demonstrated better performance than SPARK-only or spatialDE-only genes in all three different metrics: normalized Hamming distance, Jaccard distance, and Hausdorff distance ($p < 0.0001$, Kolmogorov–Smirnov test, Fig. 4h and Supplementary Fig. 20).

We further quantified the FPR of identified SV genes using the cumulative distributions of the three distances. Thresholds for false positives were determined by assuming that 95% of the commonly identified SV genes are true positives. For all four layers and all three examined distance metrics, SV genes uniquely identified by scGCO consistently demonstrated the lowest FPR (Fig. 4h and Supplementary Fig. 20). For example, 93.79% of scGCO-only genes passed the threshold derived from Jaccard distance (0.85) and could be considered as true positives in layer 2 (Fig. 4h). On the contrary, only 51.46% of SPARK-only genes and 0% of spatialDE-only genes passed the threshold in layer 2 (Fig. 4h). The favorable FPR of scGCO was also evident from directly visualizing spatial expression trends of SV genes (Supplementary Fig. 21). The high FPR was especially clear for spatialDE-only genes, where many reported spatialDE-only SV genes were only expressed in a few spots (Supplementary Fig. 21), a false positive phenomenon was previously reported by the authors of SPARK as well. Finally, the SV genes identified by scGCO are enriched with cancer and metastasis-related GO terms and pathways (Supplementary Fig. 22), confirming their biological relevance.

## ScGCO identifies SV genes from seqFISH and MERFISH data

We next analyzed the single-cell resolution seqFISH data from mouse hippocampus. The hippocampus data contains 21 different fields with variable data qualities, and consequently, the four methods identified varying number of SV genes showing large variations (Supplementary

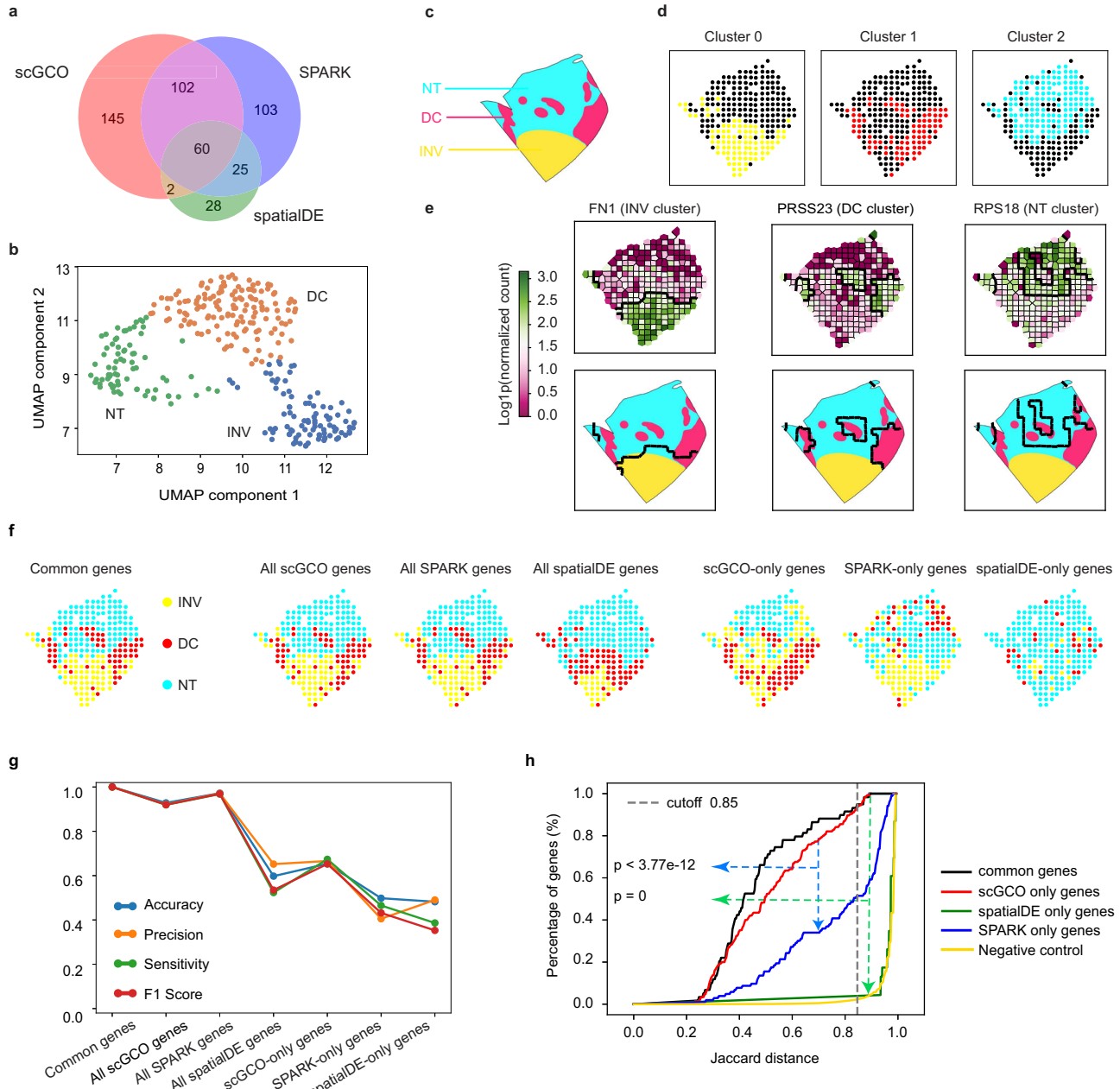

**Fig. 4 | Analyses of SV gene identification algorithms using breast cancer biopsies data. a** Venn diagram showing the gene set relationship among SV genes identified by scGCO, spatialDE and SPARK in layer 2 of breast cancer data. **b** UMAP for SV genes identified by scGCO in layer 2. **c** Annotated breast cancer tissue architecture showing the three known domains for layers 2. INV: Invasive Ductal Cancer; DC: Ductal Cancer; NT: Normal Tissue. **d** Tissue domains reconstructed with each cluster of genes identified by scGCO for layer 2. **e** Representative Voronoi diagrams showing graph cuts results (top panels), and tissue structure overlays (lower panels) for genes in (**b**). One example was shown for each cluster. Thicker black lines represent graph cuts boundaries. **f** Reconstructed tissue architectures for layer 2 using SV genes jointly identified by all three methods (first panel), with all SV genes identified by scGCO (second panel), with all SV genes identified by SPARK (third panel), with all SV genes identified by spatialDE (fourth panel), with SV genes uniquely identified by

scGCO (fifth panel), with SV genes uniquely identified by SPARK (sixth panel), and with SV genes uniquely identified by spatialDE (seventh panel) from left to right. **g** Line plots showing accuracy, precision, sensitivity and F1 score of the reconstructed tissue structures in (**f**), using the tissue structure reconstructed with common genes as reference. **h** Line plots showing the cumulative distribution of Jaccard distance of identified SV genes to the reference Breast cancer tissue structure reconstructed with common genes for layer 2. Black line: SV genes commonly identified by all three methods; red line: SV genes uniquely identified by scGCO; green line: SV genes uniquely identified by spatialDE; blue line: SV genes uniquely identified by SPARK; gold lines: genes with random expression patterns as negative control. The vertical gray dashed line indicates estimate distance cutoff corresponding to 95% of common genes. *P*-values were calculated using the two-sided pairwise KS-test without multiple-testing correction. Source data are provided as a Source Data file.

Fig. 23). The large variations suggested that the quality of seqFISH data and biological variations are dominant factors underlying the observed difference in method performance, and it is infeasible to reliable compare these methods' performance using the seqFISH data. Importantly, scGCO is the only method to robustly identify large

numbers of SV genes from all fields regardless of the quality of analyzed samples (Supplementary Fig. 23), suggesting that scGCO is an effective method to process seqFISH data under noisy conditions.

Next, we extended the analysis to MERFISH data measuring the expression of 140 genes (including 10 negative control genes) at

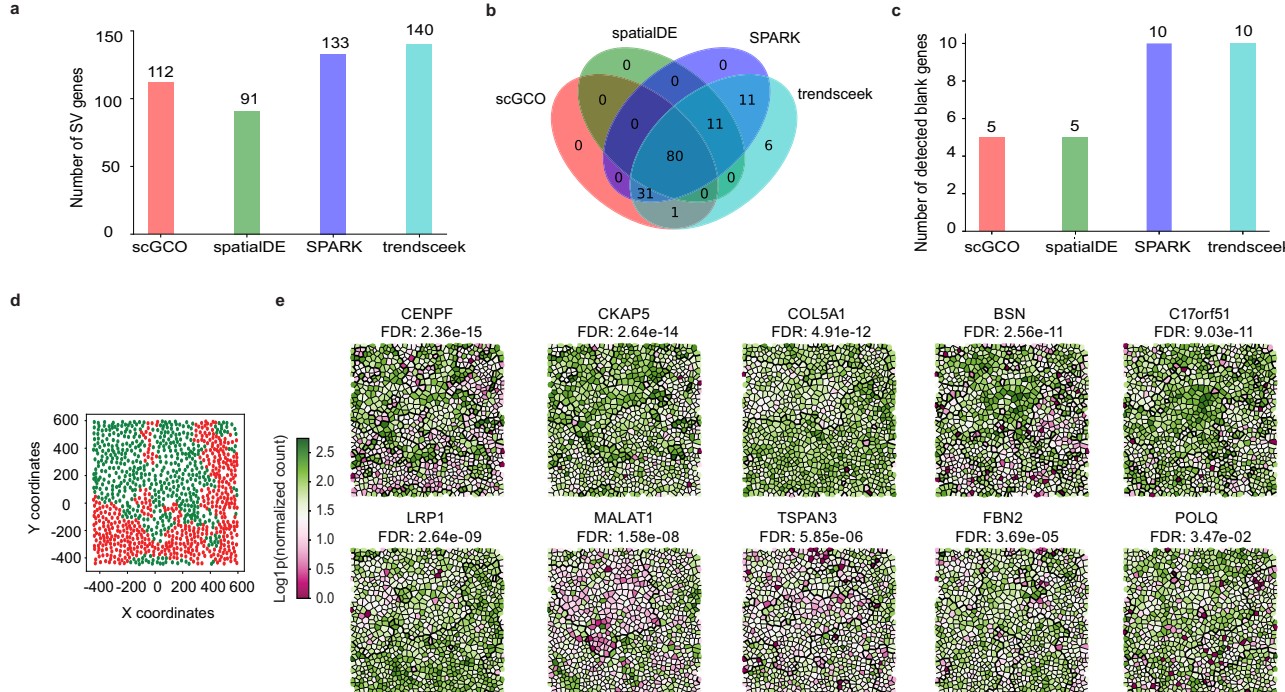

**Fig. 5 | Identification of SV genes in MERFISH data. a** Bar charts showing the number of SV genes identified by scGCO, spatialDE, SPARK and trendSceek. **b** Venn diagram showing the overlap of SV genes identified by different methods. **c** Bar charts shows the number of blank control genes identified by scGCO, spatialDE, SPARK and trendSceek as SV genes. **d** Graph cuts optimized clustering of cells showing two cell populations. **e** Graph cuts of representative SV genes identified by scGCO. Thick black lines denote the boundaries of different segments. Cells are colored according to gene expression levels.

single-cell resolution in a human cell line[4], which provides a direct measure of the FPRs of existing methods (Fig. 5). Using each method's default cutoff, scGCO identified 112 out of the 140 measured genes as SV genes including 5 negative control genes, SPARK identified 133 out of the 140 genes as SV genes including all 10 negative controls, trendSceek identified all 140 genes as SV genes including all 10 negative controls, and spatialDE identified 91 out of the 140 genes as SV genes including 5 negative controls (Fig. 5). This result confirmed that scGCO exhibits the lowest FPR among existing SV gene identification methods while still manages to identify a large number of SV genes, demonstrating the best balance of sensitivity and specificity.

## ScGCO applies to mouse neuron tissue data with LCM-seq technology

We next analyzed two mouse neuron tissue datasets using laser-capture microdissection coupled with RNA sequencing technology (LCM-Seq)[15]. The first dataset is mouse medial ganglionic eminence (MGE) (Fig. 6a–c). The resolution of the spatial transcriptome of the MGE dataset is about 50 μm, where each spot from a single tissue section was called a 'voxel', containing approximately 100 cells. The MGE dataset sequenced 24,060 genes in 127 voxels. Importantly, the MGE data set includes 96 RNA spike genes to control FPR. Consistent with simulated results (Fig. 2c), scGCO, SPARK and spatialDE all demonstrated excellent FPR, where scGCO only reported 2 spike genes as false positives and no false positive was reported by SPARK or spatialDE (Fig. 6c). In sharp contrast, SOMDE[16], a recently developed method based on self-organizing map, suffered a high false positive rate by reporting 62 out of 96 spikes as SV genes (Fig. 6c). Conversely, SOMDE reported 9,345 SV genes, while scGCO (3,867 SV genes, adjusted p-value < 0.05), SPARK (534 SV genes, adjusted p-value < 0.05) and spatialDE (72 SV genes, adjusted p-value < 0.05) reported many fewer SV genes (Fig. 6a, b). These results again suggested that scGCO demonstrates the best balance of sensitivity and specificity

by reporting a large number of SV genes while still maintaining a low FPR.

We next analyzed the expression of 28,776 genes in 101 mouse cervical spinal motor neurons (MNs) cells at postnatal day 5 (P5). This dataset achieved single-cell resolution by coupling LCM with Smart-seq2 technology[17]. ScGCO, SPARK and spatialDE reported very similar number of SV genes (7391, 7409, and 7402, adjusted p-value < 0.05), which were higher than SOMDE (3,668 SV genes) (Fig. 6d, e). Importantly, the SV genes reported by scGCO and SPARK demonstrated a highly significant overlap (73.95%, 5466 out of 7391 genes), while spatialDE uniquely reported 4367 SV genes, suggesting that scGCO and SPARK shared a higher reproducibility (Fig. 6d). Furthermore, scGCO identified all four highly expressed known MN markers *MNX1*, *CHAT*, *NEFH* and *PRPH*[28] (Fig. 6f), while the other three methods missed the key gene choline acetyl transferase (*CHAT*), which was reported to be a highly specific indicator for the functional state of cholinergic neurons in the central and peripheral nervous systems[28]. Previously, twelve homeobox transcription factors have been established to display anterior-posterior spinal cord different expression profiles[28]. ScGCO successfully identified 7 homeobox transcription factors as SV genes, while spatialDE, SPARK and SOMDE could only identify 5, 5 and 3 homeobox genes, respectively (Fig. 6g and Supplementary Fig. 24a). Furthermore, spatialDE, SPARK and SOMDE all failed to identify two significant positional expressing genes, *HOXC6* and *HOXC5*. Finally, all methods demonstrated excellent FPR by excluding non-spatial glial markers as SV genes (Fig. 6g and Supplementary Fig. 24b). Taken together, scGCO demonstrated the best combination of sensitivity and specificity on single-cell MNs spatial transcriptomics data.

## ScGCO demonstrates improved scalability
We next compared the scalability of scGCO, spatialDE, SPARK, trendSceek, and SOMDE using simulated data. We first compared the memory requirement using simulated data with cell numbers up to a million. Consistent with previous algorithm analyses results[7–9],

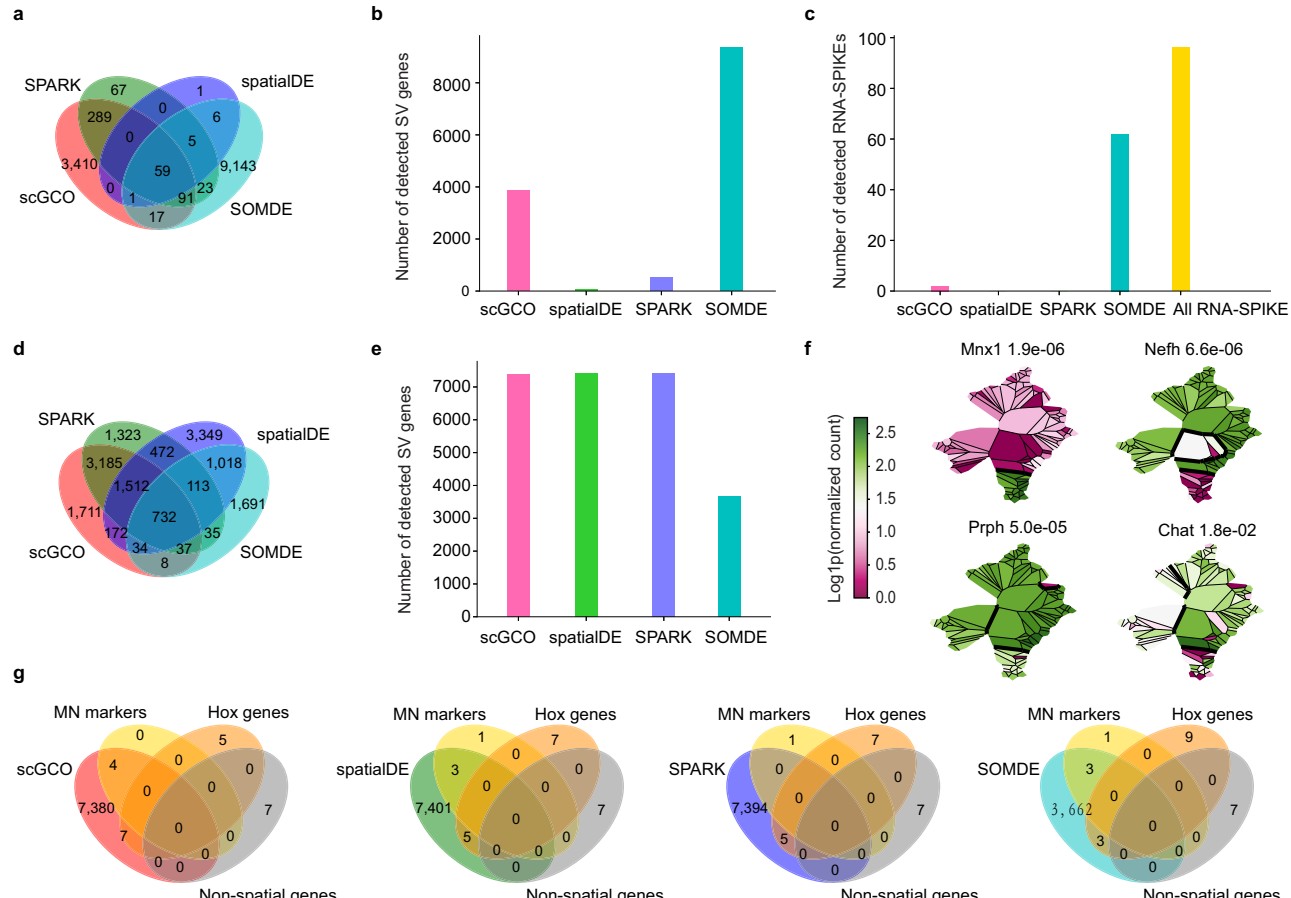

**Fig. 6 | Analyses of SV gene identification algorithms with mouse neuron tissues data generated with LCM-seq technology. a, b** Venn diagram and bar plot showing the overlap and number of SV genes identified by different methods from mouse medial ganglionic eminence (MGE). **(c)** Bar plot showing the number of detected RNA SPIKEs (designed as negative controls) by different methods. **d, e** Venn diagram and bar plot showing the overlap and number of SV genes identified by different methods from mouse motor neurons (MNs) tissues. **f** Graph cuts of representative known spatial marker genes identified by scGCO. Thick black lines denote the boundaries of different segments. Cells are colored according to gene expression levels. **g** Venn diagram showing the overlap of the MN marker genes, twelve homeobox genes, six non-spatial expression genes and SV genes identified by scGCO, spatialDE, SPARK, and SOMDE, respectively. Source data are provided as a Source Data file.

memory footprints of spatialDE, SPARK and trendSceek grow quadratically with the number of cells, and were projected to require about 100 TB, 20 TB and 20 TB memory to process 1 million cells, respectively (Fig. 7a). SOMDE demonstrated an improved memory requirement, and was projected to require 250 GB to process 1 million cells. Importantly, SPARK, spatialDE, trendSceek and SOMDE all reported memory errors while processing 1 million cells, suggesting further optimizations of these algorithms are required to process millions of cells. In contrast, owing to the sparse graph-based representation of data, scGCO's memory grows favorably with the number of cells, and can process 1 million cells using less than 8 GB memory (Fig. 7a). Similarly, the running time of spatialDE, SPARK and trendSceek is cubic or quadratic in the number of cells[7–9], and all three methods are impractical to scale to millions of cells (Fig. 7b). SOMDE demonstrated a better scalability and were projected to analyze 1 million cells in about 17.5 hours using a typical desktop computer. In contrast, scGCO's running time is near linear in the number of cells, which is consistent with reported benchmarks of graph cuts[12], and can analyze 1 million cells in about 1.5 h using a typical desktop computer (Fig. 7b).

We next evaluated these methods' scalability using several real spatial transcriptomics datasets (Supplementary Fig. 25). We first benchmarked these methods on 10 human heart tissue samples analyzed with ST sequencing, where these samples contain tens to hundreds of cells[18]. Consistent with the simulation results (Fig. 7b), scGCO

demonstrated the fastest speed among all methods. We next extended the analyses to five datasets of different mouse tissues using Slide-seq technology[19]. On these medium sized datasets containing tens of thousands of data points, scGCO demonstrated second best speed which is slightly slower than SOMDE. However, the improved performance of SOMDE was achieved by merging many cells into a single spot to improve speed. This approach sacrifices the resolution for speed, essentially defeating the purpose of single-cell technologies. Interestingly, spatialDE identified the highest number of SV genes in these datasets. A close examination revealed that the spatialDE identified SV genes have the highest percentage of empty pucks, spots where the read count is zero (Supplementary Fig. 25d). This observation suggested that some spatialDE-only genes could represent false positives, a phenomenon previously also reported by SPARK[9].

Finally, we evaluated these methods on a mega data set with MERFISH technology, including 1,027,848 cells and 161 genes[20]. ScGCO successfully processed the data in less than 2 h (115 min) using a desktop computer, a performance comparable to simulated data. Consistent with results from simulated data, the other methods couldn't process the data and reported memory errors on a server with 4 TB of memory. Taken together, these results demonstrated that scGCO demonstrates improved scalability and is a method that could process spatial transcriptomics data with millions of data points using only desktop computers.

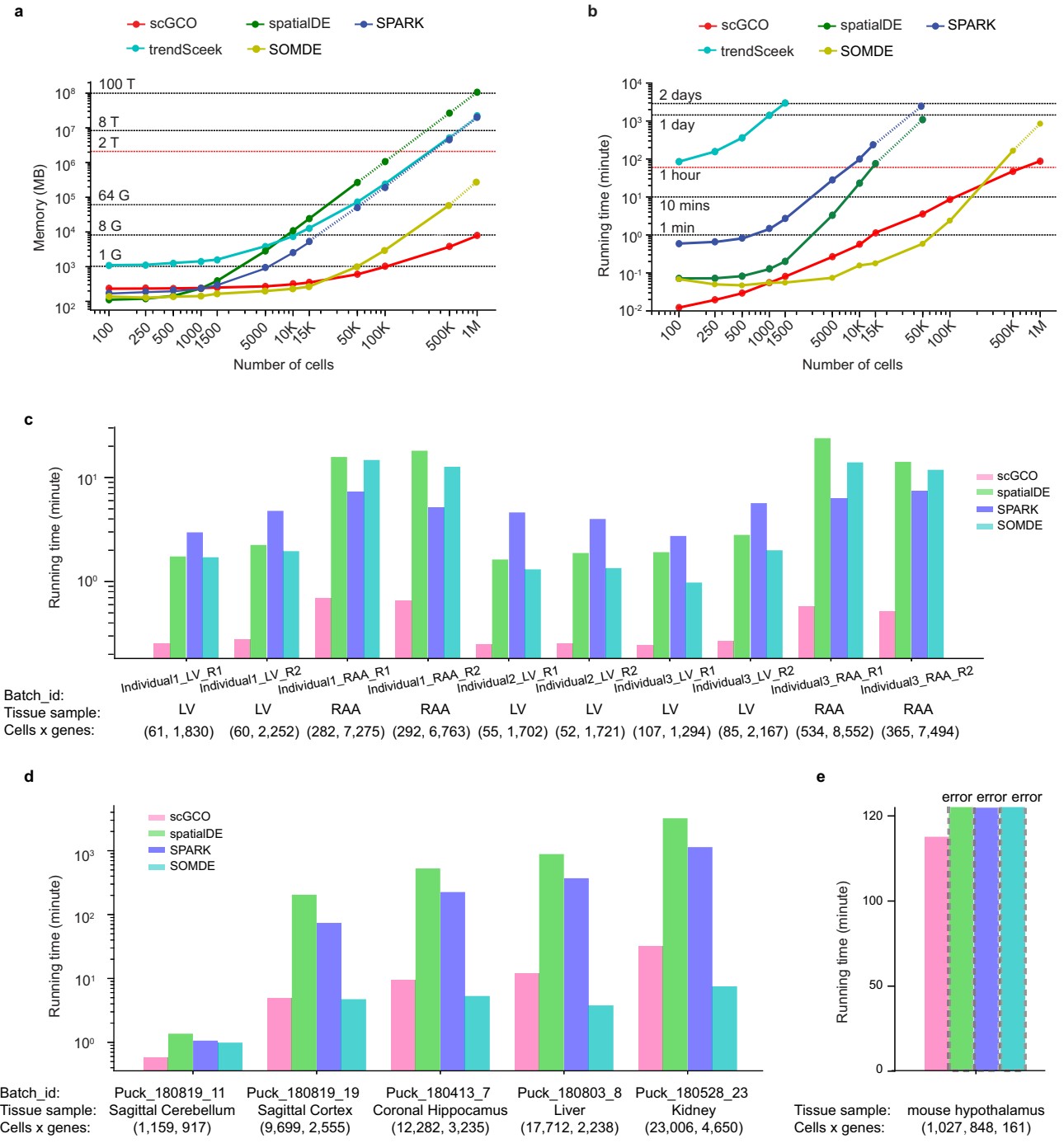

**Fig. 7 | Scalability of SV gene identification algorithms. a** Memory requirements of scGCO, SPARK, spatialDE, trendSceek, and SOMDE in the number of cells (using 100 genes) on simulated data. **b** Running time of scGCO, SPARK, spatialDE, trendSceek, and SOMDE in the number of cells (using 100 genes) on simulated data. Dotted line indicates memory or running time extrapolated from measured data. **c** Running time on adult human heart tissue data with Spatial Transcriptomic sequencing. **d** Running time on mouse complex tissues data with Slide-seq technology. **e** Running time on mouse hypothalamus data with over one million cells and hundreds of genes by MERFISH technology. Dotted bars indicate that no results were obtained due to computing errors. Source data are provided as a Source Data file.

## Discussion

Here we presented a scalable method to identify SV genes that provides a fast running time with highly robust and biologically meaningful results. Despite the favorable performance comparing to existing methods, analyses with both simulation and real biological data suggest that the performance of scGCO could still be improved. One potential source to further improve the performance of scGCO lies in the optimization procedure. Although the graph cuts algorithm is a well-established method, how to set smooth factor, the hyper-parameter of graph cuts, is an open question. A typical approach is to perform interactive graph cuts, where the user manually adjusts smooth factor based on observed results[21]. Alternatively, when a large set of annotated images are available, deep learning methods, such as neural networks, could be employed to learn the best segmentation for

each analyzed image. However, for the SV gene identification problem, where the large sets of annotated spatial patterns are unavailable, and the task is performed at genome-scale, both interactive and deep learning methods are not feasible. Here we developed an approach to identify the optimal smooth factor by maximizing signal-to-noise ratio of the graph cuts results. Because scGCO utilizes a sequential search procedure, it will likely only identify local optimal smooth factors, which could potentially lead to false positives and false negatives. Importantly, the rapid growth of spatial transcriptomics data and the progress of biological research will establish many authentic spatial patterns in the future, which could be exploited by supervised learning approaches to improve the performance of scGCO.

Single-cell sequencing technology is enjoying a rapid revolution, and data are now being generated for millions of cells in a single experiment[22]. This astronomical amount of data poses a great challenge for analysis methods, which are essential to fully realize the potentials of single-cell data. ScGCO delivers excellent scalability and can process millions of cells with desktop-level computational resources. Furthermore, scGCO also possesses an easy expandability. The graph cuts algorithm has been extended to 3-D object recognition[23] suggesting that scGCO could be readily adapted to 3-D single-cell spatial transcriptomics data. Finally, the optimization of graph cuts can be accelerated by GPU[24], and a future GPU version of scGCO could represent a valuable solution as spatial transcriptomics data grows beyond millions of cells.

## Methods

### Identifying spatial domains with HMRF-based image segmentation

Let $G = (V, E)$ be an undirected graph, where $V = v_1, v_2,…, v_N$ is the set of nodes, and $E$ is the set of edges of the graph. Let $S = \{1, 2, …, N\}$ be the set of indexes for nodes in $G$. The edges of $G$ define a neighborhood system $N = \{N_i, i \in S\}$, where $N_i$ is the set of nodes neighboring node $i$. A family of random variables $R = \{r_i, i \in S\}$ indexed by $S$, where node $i$ is associated with random variable $r_i$, is called a Markov random field if and only if $P(r_i|r_{S-\{i\}}) = P(r_i|r_{Ni})$. A hidden Markov Random Field (HMRF) is a pairwise MRF $X$ and $Y$ characterized by the following. The underlying random field $X = \{X_i, i \in S\}$ assumes values in a finite state space $L$, often called labels, whose states are not observable. The states of the emitted random field $Y = \{Y_i, i \in S\}$ are observable. Furthermore, the random variable $Y_i$ are conditionally independent given $X_i$:

$$P(\mathbf{y}|\mathbf{x}) = \prod_{i \in S} P(y_i|x_i) \quad (1)$$

Using Bayesian rules, we can express the join probability distribution of $(X, Y)$ as

$$P(\mathbf{y}, \mathbf{x}) = P(\mathbf{x}) \prod_{i \in S} P(y_i|x_i) \quad (2)$$

The central task of HMRF learning is to identify a configuration $\hat{x}$ of the hidden random field, a set of labels $X$, according to the Maximum a posteriori estimation (MAP) criterion:

$$\hat{x} = \arg \max_{\mathbf{x} \in \chi} P(\mathbf{y}|\mathbf{x})P(\mathbf{x}) \quad (3)$$

According to the Hammersley-Clifford Theorem, the joint distribution $P(\mathbf{x})$ of the hidden MRF $X$ can be equivalently characterized by a Gibbs distribution:

$$P(\mathbf{x}) = \frac{\exp(-U(\mathbf{x}))}{Z} \quad (4)$$

where $Z$ is the partition function that normalizes the distribution $P(\mathbf{x})$, and $U(\mathbf{x})$ is the energy function, $U(\mathbf{x}) = \sum_{c \in C} V_c(\mathbf{x})$, which is the sum of clique potentials over all possible cliques $C$. Similarly, the joint likelihood distribution $P(\mathbf{y}|\mathbf{x})$ can also be characterized by a Gibbs distribution:

$$P(\mathbf{y}|\mathbf{x}) = \frac{\exp(-U(\mathbf{y}|\mathbf{x}))}{Z} \quad (5)$$

Consequently, the MAP estimation of a HMRF is equivalent to minimize the posteriori energy function:

$$\hat{x} = \arg \min_{\mathbf{x} \in \chi} \{U(\mathbf{y}|\mathbf{x}) + U(\mathbf{x})\} \quad (6)$$

Studies analyzing spatial expression of genes demonstrated that the spatial distribution of expression values forms patches, where adjacent cells tend to display comparable levels of gene expression[5]. Thus, patches of cells in which a gene displays similar gene expression levels are analogous to objects in an image. Consequently, we adopted the classical energy formulation for HMRF-based image segmentation in computer vision to identify spatial domains underlying observed spatial gene expression:

$$E(X) = \sum_{p \in P} D_p(x_p) + \alpha \sum_{(p,q) \in N} V_{p,q}(x_p, x_q) \quad (7)$$

where $N$ is the set of 2-cliques (pairs of cells that interact directly) in the graphical representation of single-cell spatial gene expression data and $P$ is the set of single cells in the data set. Let $L$ be the set of possible gene expression labels determined from Gaussian mixture modeling analysis (see methods section below), then $x_p \in L$ is the label assigned to cell $p$. Consequently, $X = \{x_p|p P\}$ is the set of label assignment for all cells.

$D_p(x_p)$ is a data penalty function of assigning a particular gene expression classification label $x_p$ to cell $p$. The more likely $x_p$ is for cell $p$, the smaller is $D_p(x_p)$. Specifically, for each gene the data penalty functions for all cells can be expressed as a $P \times K$ matrix $U$, where $P$ is the number of cells and $K$ is the number of Gaussian distributions determined by Gaussian mixture modeling for the analyzed gene. The entries for $U$ are:

$$U_{p,k} = |e_p - \mu_\kappa| \quad (8)$$

where $e_p$ is the normalized gene expression at cell $p$, and $\mu_\kappa$ is the mean of the $k$th Gaussian distribution for the analyzed gene.

$V_{p,q}(x_p, x_q)$ is the interaction energy of assigning a particular pair of gene expression classifications to a pair of cells interacting directly, and $\alpha$ is the weight, also known as smooth factor, that controls the contribution of interaction energies to the overall energy. In HMRF most neighboring cells are expected to have the same label, therefore $V_{p,q}(x_p, x_q) = 0$ if $x_p = x_q$ and $V_{p,q}(x_p, x_q) > 0$ otherwise.

In order to apply graph cuts algorithm to energy minimization of HMRFs, the interaction energy must be regular[25]:

$$E^{i,j}(0,0) + E^{i,j}(1,1) \leq E^{i,j}(0,1) + E^{i,j}(1,0) \quad (9)$$

where $i$ and $j$ are indices for adjacent nodes, $E^{ij}(0,0)$ and $E^{ij}(1,1)$ represent the interaction energies when the adjacent nodes are in the same state, $E^{ij}(0,1)$ and $E^{ij}(1,0)$ represent the interaction energies when the adjacent nodes are in different states. The regularity of interaction energy guarantees a duality between energy states of HMRFs and label configurations of the corresponding graph, where the minimal energy state matches the maximum flow of the graph, hence allowing the application of graph cuts to solve energy minimization of HMRFs. In our implementation, we used a topological interaction energy that has

greater penalties when the classification of adjacent cells is further away. Specifically, the interaction energy $S$ is a symmetric matrix whose entries were:

$$S_{i,j} = |i - j|F \tag{10}$$

where $F$ is a factor that controls the magnitude of the penalty and $S_{i,j}$ is the interaction energy for adjacent cells with classification $i$ and $j$, respectively.

The goal of graph cuts optimization is to find a configuration $X$, a set of labels to all cells, that minimizes the above energy $E(X)$. When the classification of cells is limited to two classes, or two labels in an image segmentation problem, a crucial advantage of the above energy formulation of HMRFs is that powerful min-cut/max-flow algorithms for graph cuts can be used to minimize the above energy functions, which provides fast, globally optimal solutions for two-label problems[26]. For multilabel problems, global minimization of the energy function is NP-hard[11]. There are many applications of graph cuts on clustering[27]. In scGCO, we adopt the alpha-expansion algorithm developed by Boykov et al., which iteratively applies 2-label graph cuts to expand each label until the algorithm converges[11]. The algorithm runs in low polynomial time and guarantees that the solution is within a known factor of the global minimum[11].

## Converting spatial coordinates of gene expression data to graph representation

To apply the graph cuts algorithm to spatial gene expression data, we first need to represent spatial gene expression data with a HMRF graph models. We first performed Delaunay triangulation on the spatial coordinates of the cells. The graph produced by Delaunay triangulation has the nice property that only authentic neighbors are connected by edges in the graph because no cells are allowed in the triangle connecting three cells. Hence, Delaunay triangulation captures essential information of cell-cell interactions with a sparse graph. The generated graph, where each node is associated with energies derived from gene expression data and an initial label (see the next two sections in materials and methods for details), was analyzed with graph cuts algorithm to produce a segmentation. After the best segmentation has been identified by graph cuts, which represents a spatial gene expression pattern, we performed the dual operation of Delaunay triangulation to generate Voronoi diagrams, which has been broadly used to model cells[28]. To highlight the boundaries of cell clusters identified by graph cuts, edges in the Delaunay triangulation connecting cells with different predicted labels are identified, and Voronoi polygon edges intersecting these identified edges in Delaunay triangulation are highlighted, providing a direct visual representation of spatial gene expression patterns.

## Converting gene expression data to initial HMRF labels via Gaussian mixture modeling

To assign the initial labels for HMRF, we first determined a gene's expression state at each cell. We modeled the gene's log2 transformed expression values with Gaussian mixture models (GMM):

$$p(\mathbf{x}) = \sum_{\kappa=1}^{K} \pi_\kappa \mathcal{N}(\mathbf{x}|\boldsymbol{\mu}_\kappa, \boldsymbol{\Sigma}_\kappa) \tag{11}$$

where $\pi_\kappa$ is the mixing coefficient satisfying $0 \leq \pi_\kappa \leq 1$ and $\sum_{\kappa=1}^{K} \pi_\kappa = 1$; $\boldsymbol{\mu}_\kappa$ and $\boldsymbol{\Sigma}_\kappa$ are the mean vector and covariance matrix for the $k$th Gaussian distribution $\mathcal{N}(\mathbf{x}|\boldsymbol{\mu}_\kappa, \boldsymbol{\Sigma}_\kappa)$, respectively. The Gaussian mixture models were optimized with expectation–maximization (EM) algorithm. We then assigned each cell a gene expression classification according to the GMM classification of the gene's expression level in the cell. The classifications were ordered by corresponding gene expression levels so that cells with larger difference in gene expression

levels have greater difference in their classifications. This setup ensures that adjacent cells with larger expression difference are associated with larger classification differences, which will generate larger penalties in interaction energies of associated HMRFs. This energy formulation favors graph cuts that put cells with similar classifications in the same subgraph.

To determine the best number of components for GMM, we generated GMM with component numbers from 2 to 10. We then calculated Bayesian information criterion (BIC) for each GMM and selected the GMM with best BIC as final GMM for downstream analysis.

## Identification of optimal segmentation with iterative graph cuts

We developed a heuristic procedure to sequentially search for the optimal graph cuts results by varying smooth factor, the hyperparameter in graph cuts. A score quantifying the quality of a graph cuts result was defined as follows:

$$\text{Score} = -\log_{10}(P) - |\text{Noise Segments}| \times \frac{200}{n} \tag{12}$$

where $P$ is the spatial non-randomness $p$-value for the segmentation described in the below section, |Noise Segments| represents the number of noise segments (segments with $<= 9$ nodes and $p >= 0.1$) in the graph cuts result, and $n$ is the number of nodes in the graph. The second term is a normalized number of noise segments represents the average number of noise segments in a 200-nodes graph. The score essentially measures the signal-to-noise ratio of the corresponding graph cuts result.

The algorithm starts by generating the graph representation of the spatial transcriptomics data and assigns each cell an initial label as determined from GMM analysis. The search procedure then starts at smooth factor 10, with a step size of 5 or 10. At each step, we applied the alpha-expansion algorithm developed by Boykov et al., which iteratively applies 2-label graph cuts to expand each label until the algorithm converges[11]. The search stops when the score is worse than previous smooth factor, and the best graph cuts result is returned.

## Calculating the statistical significance of identified SV genes

We evaluated the statistical significance of a spatial gene expression with the complete spatial randomness (CSR) framework, where the distribution of points in 2-D plane was modeled as homogeneous spatial Poisson processes. Under the CSR model, the probability of finding exactly $k$ points of a specific label in a region $V$ can be determined from Poisson distribution:

$$P(k, \rho, V) = \frac{(V\rho)^k e^{-(V\rho)}}{k!} \tag{13}$$

where $\rho$ represents the density of the specific label, derived by dividing the number of nodes of the specific label with the total number of nodes in the HMRF.

In the context of SV genes, the HMRF consists of two sets of labels. The first set is the observed labels that are initially assigned to the nodes, which were derived from the GMM analysis of observed gene expression values, representing the observed states of gene expression. The second set is the hidden labels, which are the learned labels after the HMRF was optimized and are assumed to represent the true gene expression states of the nodes. The learned hidden labels were used to identify segments from the graph cuts, where each segment is a connected subgraph of the same learned label, presumably representing fundamental biological structures that will give rise to spatial variability in gene expression.

Once the segments were identified, we then calculated the probability that the observed gene expression states occur in each segment

using the CSR model. For each segment, $V$ is simply the number of nodes in the segment. The $k$ and $\rho$ for the analyzed segment were then derived using the observed label. Assuming the learned hidden label for the segment is $a$, then $k$ is the number of nodes in the segment whose observed label matches $a$. And $\rho$ is simply the density of $a$ calculated using the observed labels. For each candidate gene, we calculated the $p$-values for all segments identified by graph cuts and reported the best result as the spatial non-randomness $p$-value for the gene. For genome-scale analyses, multiple test correction was performed with Benjamini–Hochberg procedure.

In summary, the central task of SV gene identification is to test whether an observed gene expression distribution is dependent on spatial locations. Because the region "$V$" to exactly test for spatial non-randomness in the CSR model is generally unknown, previous method such as trendSeek can only approximate the CSR model by examining whether the distribution of the distances among all pairwise points is significantly different from randomly distributed points. The key advantage of our HMRF-based approach is to identify candidate regions exactly (these derived from HMRF-based segmentation), and directly test whether the observed gene expression displays spatial non-randomness in the candidate regions under the CSR framework.

### Benchmarking algorithms using simulated dataset

We simulated spatial gene expression datasets following the procedures described by SPARK[9]. Briefly, the spatial location (262 spots) of replicate 11 of mouse olfactory bulb data were separated into 3 groups: GCL, OPL and ONL, according to the reference tissue structures. Spots in the GCL, OPL and ONL region were assigned expression values by randomly drawing from $N(0,\sigma^2)$, $N(1,\sigma^2)$, and $N(2,\sigma^2)$, respectively. We simulated 1000 SV genes for each $\sigma$ ranging from 0.1 to 0.6 with a step size of 0.1. We also simulated 9000 non-SV genes by randomly assign expression value to each spot. Finally, we transformed the expression value to count data using the normalize_count_cellranger function in cell ranger based on the total read counts from the real data.

### Comparing spatial patterns to reference tissue structures

We used the annotated H&E staining data to generate the gold standard tissue structure, where each spot is assigned to its true layer. Specifically, we first extract the spatial coordinates of the boundaries of each layer from the annotated H&E staining. Next, the spatial coordinate of each spot is compared with the coordinates of each layer's boundary data, and each spot was assigned to the layer containing the tested spot.

We adopted the procedure described by Zhu et al[29] to reconstruct tissue structures using a set of SV genes. We first performed K-means clustering to cluster cells using selected SV genes. The k-means clustering results were set as the initial state for HMRF, and the clusters were optimized by graph cuts to generate the final optimized HMRF. Once the HMRF is optimized, the number of different labels of the HMRF is taken as the number of layers of the reconstructed tissue structure, and each spot was assigned to corresponding layer based on the HMRF label of that spot.

To calculate the normalized Hamming distance (described below), Jaccard distance, and Hausdorff distances of an SV gene to the reference tissue structures, we first match each segment of the SV gene to a layer of the reference tissue structure. A segment consists of spots with the same HMRF predicted label, and was represented by a bit vector $v$, such that $v_i = 1$ if the $i$th spot is in the segment, and $v_i = 0$ otherwise. Similarly, a layer of the reference can also be represented by a vector $u$. A segment $v$ of an SV gene is assigned to a specific layer $u$ of the reference tissue structure if $v$ and $u$ have the largest overlap among all layers of the reference structure. Once the SV gene's segments were all assigned to matching layer in the reference tissue structure, we then calculated the three metrics according to the standard definition.

Specifically, the normalized hamming distance is defined as:

$$\frac{|XOR(u,v)|_1}{|\&(u,v)|_1 + a} \quad (14)$$

where XOR represents the bitwise exclusive OR operation, and $\&$ represents the bitwise AND operation. $\|\|_1$ represents the $L_1$-norm of the vector. Finally, $a$ is a positive constant set to 10 to avoid division by zero.

### Identify highly variable genes ignoring spatial context

HVGs were identified using Seurat's FindVariableGenes function (version 2.3.4)[30]. The minimum number of genes per cell threshold was set to 200, and genes expressed in at least 3 cells were selected for downstream processing. HVGs were identified using x.cutoff 0.0125 to 3, y.cutoff 0.6, and other parameters set to default values.

### Identify genes differentially expressed between spatial domains with DESeq2

Cells in the MOB data were separated into 5 groups according to the 5 layers tissue structure resolved using SV genes identified by scGCO. Pairwise differential expression analyses were performed between all possible pairs of groups with DESeq2 using default parameters (version 1.22.2)[31]. Genes demonstrating significant differential expression were identified using FDR cutoff 0.01 and relative expression FC cutoff 1.

### Comparison to existing spatial gene identification algorithms

To systematically evaluate the performance of scGCO against published algorithms, we evaluated spatialDE (version 1.1.1), trendSceek (version 1.0.0), SPARK (version 1.0.2), and SOMDE (version 0.1.8). For spatialDE and SPARK, we downloaded the scripts provided by the authors from their GitHub website and executed the scripts without modification. For SOMDE, we installed the python package. For trendSceek, we implemented R scripts according to the methods described in trendSceek's original paper. The trendSceek's scripts and the scripts to run scGCO are provided in the tutorial files in scGCO's GitHub repository.

To evaluate these methods, we first calculated the numbers of true negatives (TN), true positives (TP), false negatives (FN), and false positives (FP). We then used these values to calculate the following four metrics: accuracy = (TP + TN)/(TP + TN + FP + FN); sensitivity = TP/(TP + FN); false positive rate (FPR) = FP/(FP + TN), and F1 score (F1) = $2*$TP/($2*$TP + FN + FP).

To estimate the scalabilities of algorithms, we evaluated memory requirement and running time using simulated data as described by Edsgard et al.[7]. For running time, we executed all algorithms on a desktop computer with Intel® Core™ i7-6700 CPU (8 cores at 3.40 GHz), 40 GiB memory, and running the Ubuntu 18.04.1 operating system. For memory profiling, we executed all algorithms on a workstation with 4 TB of memory. For spatialDE, SPARK, trendSceek, and SOMDE, these algorithms exceed the capacity of available hardware when the cell numbers are large. Because these algorithms scale quadratically or cubically with the number of cells[7–9], we estimated their memory requirement and running time by fitting available data to polynomial functions.

### Gene ontology and network analyses

The gene ontology and pathway enrichment analyses were performed using the enrichGO and enrichKEGG functions from clusterProfiler R-package (v 3.16.0)[32]. All enrichment analyses were carried out with default parameters.

### Reporting summary

Further information on research design is available in the Nature Research Reporting Summary linked to this article.

## Data availability

We downloaded the spatial transcriptomics data reported by Ståhl et al. from the Spatial Transcriptomics Research website (https://www.spatialresearch.org/resources-published-datasets/doi-10-1126science-aaf2403/)[5]. We used all 12 replicates for the mouse olfactory bulb, and all four layers for the breast cancer data. For mouse hippocampus seqFISH data[3], we downloaded the data from https://ars.els-cdn.com/content/image/1-s2.0-S0896627316307024-mmc6.xlsx. We used all 21 fields provided by the authors for analysis. The MERFISH data was downloaded from the Zhuang lab website (http://zhuang.harvard.edu/merfish.html)[4,20]. We used "Replicate 6" similar to spatialDE[8], as these had the largest number of cells and highest confluency. The LCM-seq data was downloaded from Gene Expression Omnibus (GEO) of the National Center for Biotechnology Information under the accession number GSE60402 and GSE76514[15]. The ST sequencing and slide-seq data used in this study have been available in the SpatialDB database with website: (http://www.spatialomics.org/SpatialDB/)[33]. Expression data were normalized using the same procedure as described in the cell ranger package (https://support.10xgenomics.com/single-cell-gene-expression/software/pipelines/latest/what-is-cell-ranger). Source data are provided with this paper.

## Code availability

An open-source implementation of scGCO is available at GitHub (https://github.com/WangPeng-Lab/scGCO)[34].

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

## Acknowledgements

This work was supported in part by the National Key R&D Program of China grant 2018YFC2000205, the Strategic Priority Research Program of the Chinese Academy of Sciences grant No. XDB38030100, National Science and Technology Major Project of China grant 2019ZX09201004, National Natural Science Foundation of China (NSFC) grant 31671380.

## Author contributions

K.Z. and W.F. implemented the software and performed experiments. P.W. designed the algorithm, supervised the study and implemented the software. P.W. wrote the manuscript with inputs from all authors. All authors approved the final manuscript.

## Competing interests

The authors declare no competing interests.
