## [Peer Review File · Nature Communications]

Reviewers' Comments:

Reviewer #1:

Remarks to the Author:

In this revised manuscript submitted to Nature Communications, Zhang, Feng, and Wang introduce a revised form of a previous algorithm for determining the statistical significance of spatially variable genes. They term this algorithm scGCO, and, in brief, the approach creates a graph from spatially resolved transcriptomic measurements of spots, as in Visium, or single cells, as in smFISH-based methods, then, leveraging the concept of Hidden Markov Random Fields and a null hypothesis of a completely random spatial distribution for genes, identifies the optimal cut within graphs to define spatially variable regions for individual genes as well as their statistical significance.

The authors offer a heavily revised manuscript relative to the manuscript I reviewed previously. These revisions include basic changes to their underlying algorithm to address a previous challenge with false positives as well as a more detailed description and discussion of their algorithm and its benchmarking. These benchmarks include extensive use of simulation as well as the use of scGCO (and other existing algorithms that aim to also calculate spatially variable genes) to document the computational efficiency of this algorithm and its performance as compared to other approaches. They now find that scGCO performs better or comparably in many different respects to other existing approaches though it appears to have a substantial advantage in terms of computational efficiency wrt to previous techniques.

I support the publication of this revised manuscript. However, if additional revisions are requested, I encourage the authors to consider addressing one major concern that I have as well as a series of minor concerns that I identified as well.

My only major concern revolves around the challenge of finding the optimal smooth parameter to identify when to consider the distribution of a gene across an identified spatial cut (grouped nodes on the spatial graph) as statistically distinct from the null distribution, i.e. completely random. The null distribution the authors consider strikes me as almost the proper distribution. Certainly, one should assume a completely random spatial distribution for gene expression as a part of the null hypothesis. However, when one can now group a random field on a connected graph into many different sub-graphs that are spatially connected, does the sheer number of possible combinations of different graph nodes raise the possibility that one could find a connected set of nodes that would appear to have a gene expression profile distinct from the other nodes. In other words, isn't the completely random hypothesis not sufficient, as one would need to correct this for the number of possible graph combinations being considered. The authors appear to address this concern by introducing a degree of spatial smoothing and identifying the optimal smooth parameter in an empirical fashion. However, this approach would appear to be non-ideal as it would necessarily reduce their spatial resolution to some degree.

I wonder if the authors could not comment on this situation. Is there a reason why enforcing a spatial smoothing process is the ideal solution to this problem? Or, if spatial smoothing simply reduces the number of possible graph combinations to a point where the number of combinations is not so large that it is reasonably likely to find one such combination that produces the appearance of a meaningful gene expression, could the authors describe the way in which this reduction affects their null hypothesis and the interpretation of the spatially variable genes they find?

In addition, I have a series of quite minor concerns that I list here just to bring them to the attention of the authors in the hope that they might help strengthen the manuscript.

First, the authors cite a general review on the importance of spatial transcriptomics: Martin & Ephrussi Cell 2009. But, by my reading, this review is more on the importance of the internal organization of RNAs within cells. There are more recent reviews that could represent better references for the field of spatial transcriptomics, in general.

Second, throughout the authors utilize language to describe their work that may be a bit too aggressive. Terms such as 'unparalleled' and the like may not be necessary or helpful.

Third, could the authors formally define terms such as accuracy and sensitivity when they use them?

Fourth, the authors adopt three characteristics as the definition of spatially variable genes: spatial dependence on gene expression, conserved spatial patterns, and agreement with underlying tissue structure. However, I would argue that only the first point is truly relevant to the most general definition of spatially variable genes and that there will be biological instances in which spatially varying gene expression will not be reflected in the underlying tissue structure or not be mapped to reproducible tissue configurations. If the authors feel these additional constraints are necessary, perhaps, they could explain why in a few more words?

Finally, the authors test the computational capabilities of scGCO as well as other existing algorithms on a variety of datasets, e.g. slide-seq or MERFISH data, yet they do not present any of the results of these algorithms. I wonder if it might not be worth at least a supplemental figure describing what they observed in terms of spatially variable genes and their patterns in these data as well.

Reviewers' comments:

Reviewer #1:

Remarks to the Author:

My only major concern revolves around the challenge of finding the optimal smooth parameter to identify when to consider the distribution of a gene across an identified spatial cut (grouped nodes on the spatial graph) as statistically distinct from the null distribution, i.e. completely random. The null distribution the authors consider strikes me as almost the proper distribution. Certainly, one should assume a completely random spatial distribution for gene expression as a part of the null hypothesis. However, when one can now group a random field on a connected graph into many different sub-graphs that are spatially connected, does the sheer number of possible combinations of different graph nodes raise the possibility that one could find a connected set of nodes that would appear to have a gene expression profile distinct from the other nodes. In other words, isn't the completely random hypothesis not sufficient, as one would need to correct this for the number of possible graph combinations being considered. The authors appear to address this concern by introducing a degree of spatial smoothing and identifying the optimal smooth parameter in an empirical fashion. However, this approach would appear to be non-ideal as it would necessarily reduce their spatial resolution to some degree.

I wonder if the authors could not comment on this situation. Is there a reason why enforcing a spatial smoothing process is the ideal solution to this problem? Or, if spatial smoothing simply reduces the number of possible graph combinations to a point where the number of combinations is not so large that it is reasonably likely to find one such combination that produces the appearance of a meaningful gene expression, could the authors describe the way in which this reduction affects their null hypothesis and the interpretation of the spatially variable genes they find?

Response: we thank the reviewer for raising this very insightful question. We will first discuss the rationale of using complete randomness as the null hypothesis, followed by a discussion on the impact of using smoothness process on the null hypothesis and the identification of SV genes.

Considering points on a 2D plane, there are essentially three types of relationships among these points: positively correlated, negatively correlated, and not correlated. For positively correlated points, they will have a strong tendency to form clusters. Undirected graph models such as HMM are designed to describe such positive correlations and to identify clusters. For negatively correlated points, they repel each other and have a strong tendency to not form clusters. These properties have been utilized in recommendation systems, and models such as determinantal point process have been developed to capture such negative correlations. Given that both positively correlated, and negatively correlated points can generate strong non-random patterns, it is natural to use the middle ground, where points are not correlated, also known as complete randomness, as the null hypothesis. The null hypothesis based on positively

correlated points is biased towards high false negative rate because only strong clusters will be considered statistically different from background; And the null hypothesis based on negatively correlated points is biased towards high false positive rate because even weak random clusters will be considered statistically different from background. Hence, complete randomness appears to be the unbiased natural choice as the null hypothesis.

An undesirable property of randomly distributed points is that they do be able to form clusters, albeit at a much lower probability than positively correlated points. Normally, this can be well-controlled using FDR based cutoffs. Unfortunately, as we discussed in our previous response to reviewer's comments, the limited size of the region examined by spatial transcriptomics technologies and the large number of cells/spots that will overexpress a specific gene in the examined region, could create many more connected components (sub-networks) than in a large region with fewer cells/spots overexpressing the tested gene, and simply using FDR based cutoff to reduce these false positive genes will leads to elevated false negative rate.

To combat this issue, we took an inductive reasoning approach by examine the properties of authentic SV genes and randomly generated spatial patterns. These analyses suggested that authentic SV genes appears to be very robust against increasingly larger smooth factors, while the randomly generated spatial patterns are not. We then reasoned that we could generalize the robustness property to all SV genes and designed the signal-to-noise-ratio based approach, where we search for an "optimal" smooth factor to help to identify authentic SV genes.

Thus, from empirical evidence that authentic SV genes are highly robust and resistance to increasing smooth factors, our procedure represents a good balance of strongly reducing false positives without a large impact on wrongly reducing true positives.

However, given that biological systems are highly dynamic, we will not be surprised that SV genes could display highly dynamic spatial patterns that changes during development, disease and response to internal/external stimulus. Thus, if authentic SV genes could indeed display dynamic spatial patterns that are sensitive to the change of smooth factors, our smooth procedure will "inflate" the null hypothesis and leads to high false negative rates. Consequently, the spatially variable genes identified by scGCO will bias toward the steady state SV genes that are resistant to the change of smooth factors, and scGCO could perform less well on identifying dynamic SV genes whose spatial patterns are sensitive to the change of smooth factors.

We thank again for the reviewer to raise this insightful point. We will continue to monitor the progress in spatially variable gene research, and will continue to exploit newly discovered properties of SV genes to improve scGCO.

In addition, I have a series of quite minor concerns that I list here just to bring them to

the attention of the authors in the hope that they might help strengthen the manuscript.

First, the authors cite a general review on the importance of spatial transcriptomics: Martin & Ephrussi Cell 2009. But, by my reading, this review is more on the importance of the internal organization of RNAs within cells. There are more recent reviews that could represent better references for the field of spatial transcriptomics, in general.

Response: we have updated the references to more recent and more representative ones as suggested.

Second, throughout the authors utilize language to describe their work that may be a bit too aggressive. Terms such as ‘unparalleled’ and the like may not be necessary or helpful.

Response: we thank reviewer for pointing out this problem and have toned down these descriptions to use more conservative terms.

Third, could the authors formally define terms such as accuracy and sensitivity when they use them?

Response: we have added definition of these metrics in the methods section “**Comparison to existing spatial gene identification algorithms**”.

Fourth, the authors adopt three characteristics as the definition of spatially variable genes: spatial dependence on gene expression, conserved spatial patterns, and agreement with underlying tissue structure. However, I would argue that only the first point is truly relevant to the most general definition of spatially variable genes and that there will be biological instances in which spatially varying gene expression will not be reflected in the underlying tissue structure or not be mapped to reproducible tissue configurations. If the authors feel these additional constraints are necessary, perhaps, they could explain why in a few more words?

Response: we thank reviewer for pointing out this caveat and have updated the manuscript according to the reviewer’s suggestions. Specifically, the SV gene now only defined by the first point. And we followed the definition by the other two points, suggesting that these two points are often observed and are generally associated with SV genes, instead of being required as in our original definition.

Finally, the authors test the computational capabilities of scGCO as well as other existing algorithms on a variety of datasets, e.g. slide-seq or MERFISH data, yet they do not present any of the results of these algorithms. I wonder if it might not be worth at least a supplemental figure describing what they observed in terms of spatially variable

genes and their patterns in these data as well.

Response: we thank reviewer for this important suggestion and have added results from these data sets as supplementary figure 25. We also included a brief discussion of these results in the main text in the section “**ScGCO demonstrates improved scalability**”.